# Label Free Glucose Electrochemical Biosensor Based on Poly(3,4-ethylenedioxy thiophene):Polystyrene Sulfonate/Titanium Carbide/Graphene Quantum Dots

**DOI:** 10.3390/bios11080267

**Published:** 2021-08-07

**Authors:** Siti Nur AshakirinMohd Nashruddin, Jaafar Abdullah, Muhammad Aniq Shazni Mohammad Haniff, Mohd Hazani Mat Zaid, Ooi Poh Choon, Mohd Farhanulhakim Mohd Razip Wee

**Affiliations:** 1Institute of Microengineering and Nanoelectronics (IMEN), Universiti Kebangsaan Malaysia (UKM), Bangi 43600, Selangor, Malaysia; p103435@siswa.ukm.edu.my (S.N.A.N.); aniqshazni@ukm.edu.my (M.A.S.M.H.); pcooi@ukm.edu.my (O.P.C.); 2Department of Chemistry, Faculty of Science, University Putra Malaysia, Serdang 43400, Selangor, Malaysia; jafar@upm.edu.my

**Keywords:** screen printed carbon electrode, nanocomposite, biosensor, glucose oxidase, electrochemical biosensor, polystyrene sulfonate (PEDOT:PSS), titanium carbide (Ti_3_C_2_), graphene quantum dots (GQD), redox mediator free

## Abstract

The electrochemical biosensor devices based on enzymes for monitoring biochemical substances are still considered attractive. We investigated the immobilization of glucose oxidase (GOx) on a new composite nanomaterial poly(3,4-ethylenedioxythiophene): polystyrene sulfonate (PEDOT:PSS)/titanium carbide,(Ti_3_C_2_)/graphene quantum dots(GQD) modified screen-printed carbon electrode (SPCE) for glucose sensing. The characterization and electrochemical behavior of PEDOT:PSS/Ti_3_C_2_/GQD towards the electrocatalytic oxidation of GOx was analyzed by FTIR, XPS, SEM, cyclic voltammetry (CV), and differential pulse voltammetry (DPV). This composite nanomaterial was found to tend to increase the electrochemical behavior and led to a higher peak current of 100.17 µA compared to 82.01 µA and 95.04 µA for PEDOT:PSS and PEDOT:PSS/Ti_3_C_2_ alone. Moreover, the detection results demonstrated that the fabricated biosensor had a linear voltammetry response in the glucose concentration range 0–500 µM with a relatively sensitivity of 21.64 µAmM^−1^cm^−2^ and a detection limit of 65 µM (S/N = 3), with good stability and selectivity. This finding could be useful as applicable guidance for the modification screen printed carbon (SPCE) electrodes focused on composite PEDOT:PSS/Ti_3_C_2_/GQD for efficient detection using an enzyme-based biosensor.

## 1. Introduction

Diabetes disease is a condition with an excess of blood glucose that mainly originates from unhealthy food consumption [1]. Thus, without continuous careful management, it can lead to the build-up of sugar in the blood, which can raise the risk of harmful complications, including strokes and heart disease. According to the World Health Organization (WHO) report, diabetes was the seventh leading cause of death in 2016. Experts also claim that diabetes can be reversed as early as 3–5 years in disease by normalizing blood sugar levels without medication. Therefore, early identification of glucose in the blood would prevent diabetes [2], presenting a need to establish simple diagnoses for potential patients with diabetes.

The electrochemical biosensor has become an attractive choice for high throughput analysis with increased sensitivity, rapidity, and selectivity of data collection by using a small volume of the sample [3]. Moreover, in advance of technology, the screen-printed electrode seems widely used in the electrochemical research field due to its simplicity, low cost, simple fabrication, fast response, disposable, portable, and can be used as a point of care (POC) device [4]. Nevertheless, the limitations of screen-printed carbon electrodes (SPCE) are mostly related to low current if used without any modifications which restrict its selectivity and sensitivity. To increase the efficiency of SPCE, the working electrode surface normally needs to be modified by adding a modifier that provides higher conductivity and accelerates the transfer of electrons [5]. Such electrode modifications may strengthen signal transduction at the electrode–electrolyte interchange by improving sensitivity and limits of detection [6].

The advent of conductive polymers such as PEDOT:PSS, poly(3,4-ethylenedioxythiophene) polystyrene sulfonate has been used to enhance the conductivity of electrodes with moderate band gap, good stability, and low redox potential [7]. It can be easily deposited as thin films by various methods such as spin-coating, dip-coating, or inkjet micro deposition [8]. In addition, it is highly employed in enzyme-based biosensors due to its high enzyme immobilization ability [9]. However, the poor electrocatalytic capability of PEDOT:PSS limited its electrochemical application in sensors [10]. Few studies have demonstrated the incorporation of graphene-based material or metallic nanoparticles into PEDOT:PSS to form a nanocomposite that stabilizes the electron transport and improves its flexibility and conductivity [11,12]

Titanium carbide, Ti_3_C_2_, also called MXene, an evolving family of (2D) materials inorganic materials, holds great promise as the ideal candidate sensing platform to produce electrochemical biosensors have shown outstanding electronic properties [13]. MXene based composite materials have received particular attention due to the synergistic effect, low interfacial resistance, and electrical conductivities [14]. Previously, a PEDOT:PSS/Ti_3_C_2_ hybrid composite demonstrated improved electrochemical sensing capabilities and showed excellent conductivity compared to the simple PEDOT:PSS modified electrodes [15].

Among graphene families, graphene quantum dots demonstrate high improvement due to quantum confinement and edge effects, giving rise to the unique electronic, optoelectronic, photoelectric, enlarged surface to mass ratio, and high conductivity properties [16]. Graphene quantum dot (GQD) has zero-dimensional (0D) nanostructure derived from the carbon family with properties derived from both graphene and carbon dots (CDs) [17]. GQD is a non-toxic and bio-inert material that is considered environmentally friendly [18,19]. In addition, GQD also can be dispersed in water easily due to its functional groups providing hydrophilicity conditions [20]. The electrochemical behavior and electrical conductivity of composite materials such as PEDOT:PSS/Ti_3_C_2_ can be improved by GQD modification [21]. The PEDOT:PSS/Ti_3_C_2_/GQD nanocomposite modified electrodes were used without a binder to increase the electrochemical reaction rate, reduce adsorption, and shorten the ion diffusion path.

In this nanocomposite, PEDOT:PSS will act as an effective electrical conductor as well as a polymeric binder to produce a stable dispersion of GQD and Ti_3_C_2_, which are important in improving electrocatalytic activity towards glucose detection [22]. Therefore, we proposed PEDOT:PSS/Ti_3_C_2_/GQD for electrode modification. The variable concentration of glucose as an analyte of study was analyzed using the differential pulse voltammetry (DPV) technique. Moreover, the stability, repeatability, and interference effects of the proposed biosensor have also been investigated. Using facile methods with the involvement of any cross-linker, it was found that the proposed biosensor was sensitive enough for the detection of glucose with excellent selectivity and stability.

## 2. Materials and Methods

### 2.1. Chemical and Reagents

Glucose, glucose oxidase (GOx), and poly(3,4-ethylenedioxythiophene)-polystyrenesulfonate (PEDOT:PSS, 666201) were purchased from Sigma-Aldrich, St. Louis, MO, USA. Graphene quantum dot (GNQD0101) was purchased from ACS Materials and titanium carbide, Ti_3_C_2_ was derived from the typical exfoliation of MAX phase (Ti_3_AlC_2_) method [23]. Acetic acid and ammonia were purchased from Sigma-Aldrich were mixed to obtain a solution of 0.1 M ammonium acetate solution, pH 7. Potassium ferricyanide K_3_Fe(CN)_6_, was purchased from R&M Chemicals, Malaysia. Deionized (DI) water was used throughout the experiments.

### 2.2. Instrumentation

Cyclic voltammetry and DPV were performed using mini PSTAT 910 (Metrohm Ltd. Herisau, Switzerland) controlled by the personal computer. A screen-printed carbon electrode (SPCE, DRP C110) with a diameter of 4 mm was purchased from Dropsens (Oviedo, Spain). The electrode consists of the working electrode (carbon), reference electrode (Ag/Cl), and counter/auxiliary electrode (carbon), respectively.

### 2.3. Preparation of PEDOT:PSS/Ti_3_C_2_/GQD-GOx Modified Electrode

Preparation of PEDOT:PSS/Ti_3_C_2_/GQD of 1:1:1 (*v*/*v*/*v*) ratio was mixed and sonicated at 30 °C for 10 min. Then, the nanocomposite material was stored overnight (24 h) at room temperature. Subsequently, 3 µL of PEDOT:PSS/Ti_3_C_2_/GQD nanocomposite was uniformly drop-cast onto the electrode surface of the SPCE and left to dry (2 h) at room temperature. Prior to the modification phase, the electrode is activated by CV for approximately 50 cycles in KCl at a scan rate of 100 mV/s. Next, 3 µL of 0.1 M GOx was immobilized on the modified SPCE and left for two hours to dry. After the enzymatic layer was formed, the electrode was rinsed with a stream of a phosphate-buffered solution to remove the residual monomer and weakly linked enzyme molecule prior to taking the measurements. A schematic diagram of PEDOT:PSS/Ti_3_C_2_/GQD/GOx modified SPCE glucose biosensor is illustrated in Figure 1. The detection mechanism of the developed biosensor in this study can be classified as a third-generation glucose biosensor which is capable of direct electron transfer, providing an electrical signal to be measured.

### 2.4. Characterization

A scanning electron microscope (SEM) (JEOL) was used to observe the surface morphologies of nanocomposites on SPCE. A high-resolution transmission electron microscope (HR-TEM) system (JEOL, JEM-2010) operated at an accelerating voltage of 200 kV was used to determine the size and distribution of the nanomaterials. FTIR spectroscopy (FT/IR-6100TypeA, JASCO) was carried out to examine the functional group of nanocomposite materials with immobilized glucose oxidase. X-ray photoelectron spectroscopy (XPS) on an Axis Ultra DLD, Kratos/Shimadzu with Al monochromatic source (1486.69 eV) with charge compensation system was utilized to determine the elemental composition and oxidation state of the prepared samples. The obtained results were analyzed by CasaXPS software using the Shirley algorithm and GL (30) line shape.

### 2.5. Electrochemical Measurements of PEDOT:PSS/Ti_3_C_2_/GQD-GOx

The calibration curves were statistically analyzed by OriginPro 8.0 and the relevant results (slope and intercept) were evaluated. All voltammetric measurements were performed in triplicate (*n* = 3) at room temperature. Limit of detection (LOD) was calculated based on three times the standard deviation of the intercept divided by the slope of the calibration curve. The effective surface area of the electrode was determined by the Randles–Sevcik equation:*i*_p_ = (2.69 × 10^5^)n^3/2^*D*^1/2^C*A*v^1/2^(1)
where the *n* is the number of transferred electrons for the redox reaction, *D* is the diffusion coefficient (6.7 × 10^−6^ cm^2^s^−2^), C is the molarity of ferricyanide 0.1 M, *A* is the effective surface area in cm^2^, and *v* is the scan rate (mVs^−1^).

The effective surface area was performed at various scan rates, and with a well-established linear relationship between current (*i*) and square root scan rate (*v*^1/2^) by performing linear regression for *i* versus *v*^1/2^, the slope *k* was obtained and the effective surface area [24] is expressed as:(2)$$A=\frac{k}{\left(2.69\times 10^{5}\right)n^{3/2}D^{1/2}C}$$

## 3. Results

### 3.1. FTIR and XPS Analysis

The FTIR spectra of PEDOT:PSS/Ti_3_C_2_/GQD were measured in the presence and absence of glucose oxidase immobilization. Figure 2A (spectrum a) indicates the transmittance of the PEDOT:PSS/Ti_3_C_2_/GQD without GOx displays a weak C≡C band at a peak of 2116 cm^−1^, while the presence of a peak at 1640 cm^−1^ is assigned to the C=C group. The absorption peak at 1513 cm^−1^ shows C=O stretching vibration, which suggests the presence of oxygen-containing groups in PEDOT:PSS. Moreover, an exhibition of several fingerprint bands due to the existence of PEDOT:PSS for C-O and C-O-C stretching mode (ethylenedioxy group) at a peak of 1292 cm^−1^ and 1025 cm^−1^, respectively. Other bands were formed at a peak of 899 cm^−1^ and 755 cm^−1^ due to C-H aromatic bending. Subsequently, spectrum b containing the predicted bands of PEDOT:PSS and GOx functional groups confirms the active immobilization of the enzyme. The existence of carboxyl-activated PEDOT:PSS can be determined by the broad peaks at 3282 cm^−1^ and 2950 cm^−1^ corresponding to the O-H and C-H stretch, with the another peak at 1637 cm^−1^ that represents the C=C stretch. In addition, the sulphate group in the PEDOT:PSS backbone contributes to the C-O and S=O stretching bands at 1187 cm^−1^ and 1045 cm^−1^. Furthermore, PEDOT:PSS/Ti_3_C_2_/GQD indicates more pronounced peaks from the GOx group. Thus, GQD can also help boost the immobilization of GOx on SPCE [18]. Moreover, Ti_3_C_2_ induces perturbations of certain peak amplitudes and locations as seen at peaks 598 cm^−1^ and 627 cm^−1^ (Ti-O) for modified electrodes immobilized with GOx and without GOx [18].

Subsequently, the chemical constitution and the structure of the PEDOT:PSS/Ti_3_C_2_/GQD were investigated by X-ray photoelectron spectroscopy (XPS). The general survey of the XPS spectrum of the PEDOT:PSS/Ti_3_C_2_/GQD shows three strong binding energy peaks at 165.4, 284.9 and 530.6 eV were attributed to S 2p, C 1s, and O 1s, respectively (S1). The PEDOT:PSS/Ti_3_C_2_/GQD were composed of S (1.56%), C (78.81%), and O (19.63%), especially, in the high-resolution spectrum of S 2p (Figure 2B), where there were two peaks at 160.6 (0.09%) and 161.6 (0.49%) which were attributed to S 2p^3/2^, and one peak at 165.4 (1.03%) eV which was attributed to S 2p^1/2^ species, respectively. Three remarkable peaks in the C 1s spectrum at 284.9 (58.26%), 286.5 (13.32%), and 289.0 (5.56%) eV (Figure 2C) were attributed to C-C, C-O, and C=O groups. In the O 1s spectrum (Figure 2D), two peaks at 529.2 (13.61%) and 530.6 (7.63%) eV both were attributed to O-Ti groups, indicating the presence of Ti bonding by MXene [25,26].

Subsequently, Figure 3 shows HRTEM and SEM characterization was carried out to investigate the morphology of nanocomposite thoroughly. As depicted in Figure 3A, the GQD morphology was investigated with a high-resolution transmission electron microscope (HRTEM) display a quasi-spherical, mono-dispersed, and homogeneously distributed with an average diameter of less than 10 nm. Further, the morphology and structure of Ti_3_C_2_ and the composite nanomaterial (PEDOT:PSS/Ti_3_C_2_/GQD) were then analyzed by using SEM as shown in Figure 3B,C. It shows the top view of images of Ti_3_C_2_ display two-dimensional multilayers and restacking in the sheet which in agreement with the previous report [13,14]. Subsequently, the nanocomposite shows a more aggregated, rough surface, resembling flakes, probably due to high surface energy between Ti_3_C_2_ and GQD nanoparticles which resulted in agglomeration [27].

### 3.2. Electrochemical Characterization

Figure 4A shows the cyclic voltammogram of PEDOT:PSS, PEDOT:PSS/Ti_3_C_2,_ and PEDOT:PSS/Ti_3_C_2_/GQD, respectively in the presence of K_3_[Fe(CN)_6_] as a redox probe. A pair of well-defined redox peak is observed with a peak to peak separation (ΔEp) of 140 mV for bare SPCE, whereas the ΔEp exhibited by the PEDOT:PSS, PEDOT:PSS/Ti_3_C_2_, and PEDOT:PSS/Ti_3_C_2_/GQD is about 210, 230, and 234 Mv and respectively. When compared to the bare SPCE, the redox peak current demonstrated a significant increase for the PEDOT:PSS/Ti_3_C_2_/GQD.

In addition, the effect of scan rates (from 25 to 150 mVs^−1^) on the peak currents at PEDOT:PSS/Ti_3_C_2_/GQD electrode was investigated. As can be seen in Figure 4B–D, the plot of the square root scan rate vs. peak current of bare SPCE, PEDOT:PSS/Ti_3_C_2_, and PEDOT:PSS/Ti_3_C_2_/GQD (inset) exhibited a linear calibration with coefficient regression of 0.9999, 0.9895, and 0.9953 respectively, showed that the process is diffusion-controlled [28]. Moreover, by using the Randles–Sevcik equation, the calculated effective surface area for bare SPCE, PEDOT:PSS/Ti_3_C_2_, and PEDOT:PSS/Ti_3_C_2_/GQD were 0.013 cm^2^, 0.021 cm^2^, and 0.026 cm^2^, respectively. Compared to bare SPCE, the effective surface area (A) of PEDOT:PSS/Ti_3_C_2_ was significantly enhanced by about 62% compared to bare SPCE. This suggested that the increment of the electroactive surface was largely attributed to PEDOT:PSS/Ti_3_C_2._

### 3.3. Response Characteristics of the PEDOT:PSS/Ti_3_C_2_/GQD-GOx Electrode toward Glucose

The CVs of PEDOT:PSS/Ti_3_C_2_/GQD electrode in 0.1 M ammonium acetate solution (pH = 7.0) in the presence and absence of 500 µM glucose oxide are recorded in Figure 5. It can be seen that a small oxidation and reduction peak was observed after the addition of GOx, which can be attributed to the consumption of O_2_ in ammonium acetate solution, which resulted in the redox process [29].

### 3.4. Optimal pH of PEDOT:PSS/Ti_3_C_2_/GQD

In order to examine the idyllic pH for glucose detection at PEDOT:PSS/Ti_3_C_2_/GQD SPCE, different pH buffer solutions (pH 4.5–pH 8) have been investigated using DPV and CV, respectively [30]. Figure 6A shows the influence of pH on the developed biosensor, the results reveal that pH 7.0 has the highest peak oxidation current suggesting the optimum for enzyme activity in this study [30]. In addition, above pH 8.0, the oxidation peak current decreased significantly which might be due to denaturation and a loss in enzyme activity [31]. Therefore, the ammonium acetate solution at pH 7.0 was selected for further investigation of PEDOT:PSS/Ti_3_C_2_/GQD/GOx SPCE. Moreover, in most cases, the redox behaviors of GOx are often greatly dependent on the solution pH. In this study, a negative shift of both the redox peak potentials of GOx appeared when the pH value of the solution increased. The formal potential (E0′) showed a good linear function with pH solution in the range of 4.5–8.0 where the linearity for glucose detection has been confirmed by the peak potential plot in Figure 6B. The acquired slope value was found to be 58.1 mV per pH unit, following the Nernst equation [32], which confirms the same number of protons and electrons involved in the PEDOT:PSS/Ti_3_C_2_/GQD SPCE mechanism [33].

### 3.5. PEDOT:PSS/Ti_3_C_2_/GQD/SPCE as a Transducer

Figure 7 depicts the correlation between sensor responses and target concentrations using differential pulse voltammetry and ferrocyanide/ferricyanide redox probe. Using this platform, the limit of detection (LOD) and limit of quantification (LOQ) were calculated using 3σ/slope and 10σ/slope respectively, where σ is the residual standard deviation. As a result, PEDOT:PSS/Ti_3_C_2_/GQD modified SPCE showed an outstanding performance for glucose determination in the broad linear range 0–500 µM (R^2^ = 0.9875) with the limit of detection (LOD) and limit of quantification (LOQ) of 65 µM and 217 µM (Figure 7A,B) respectively. Meanwhile, the calculated sensitivity exhibited high sensitivity of about 21.64 µAmM^−1^cm^−2^, where the values seemed to be within an acceptable range and [34] comparable to the previous study, as shown in Table 1. The linearity of glucose concentration on modified electrodes (Figure 7B) with regression value is almost 1.0. 

### 3.6. Stability, Repeatability and Selectivity of the Modified Electrode

Repeatability studies of the developed sensor were measured using cyclic voltammetry (CV). A concentration of 500 µM glucose demonstrates a good current signal with 4 consecutive cycles using the same electrode with a relative standard deviation (RSD) of 2.05%. The steady-state current was still sustained at 90% of its initial response (Figure 8A). This means that a slight change in the oxidation-reduction peak was observed (Appendix A). The repeatability values are considered acceptable for four replications [41]. The long-term storage stability of PEDOT:PSS/Ti_3_C_2_/GQD with GOx on SPCE was also investigated. After one-month storage at ambient conditions, a slight decrease in the anodic and cathodic peak current [42] was observed (Figure 8A). The anodic and cathodic peak current was still retained above 81% and 72% of its initial response after one month (Figure 8A). The result shows that GOx is not leaching on the electrode surface even after 1 month. This suggested that the fabricated biosensors had outstanding storage stability and durability for analyte sensing in the free solution.

The interference effect analysis on glucose determination was performed using DPV under the same experimental conditions (Figure 8B). The interference of glucose with the presence of AA, UA, and creatinine was investigated using 50 M, and the effects on the current response of the proposed biosensor are shown in Figure 8B. During the measurement, the oxidation currents of AA and creatinine have been observed, as a result, no significant response was observed for the addition of ascorbic acid (0.25%), uric acid (0.05%) and creatinine (0.18%). These findings indicate that SPCE/PEDOT:PSS/Ti_3_C_2_/GQD/GOx is highly selective for glucose and had no interference with AA, UA, and creatinine. 

## 4. Discussion

In this study, GOx was immobilized only by physical trapping which is governed by a noncovalent interaction strategy, GOx immobilization with natural bonding like hydrophobic, hydrogen bonds, and pi-stacking on the modified electrode is expected. Thus, FTIR characterization has shown appropriated peaks have appeared for both spectra (Figure 2A,B). This peak was responsible for helps GOx immobilization on PEDOT:PSS/Ti_3_C_2_/GQD. For XPS characterization, the results obtained indicated that the PEDOT:PSS/Ti_3_C_2_/GQD were composed of three elements of C, S, and O with multiple oxygen-containing groups and titanium groups on the surface of the PEDOT:PSS/Ti_3_C_2_/GQD. Due to the strictly superficial nature of the XPS analysis, there is clear evidence of the presence of GOx protein-covered elements [26].

As far as morphologies are concerned, GQD and Ti_3_C_2_ showed the expected images where the findings were satisfied by previous studies [13,14]. Meanwhile, the images for composite nanomaterial show a more aggregated rough surface, which might due to the high surface energy between Ti_3_C_2_ and GQD nanoparticles that resulted in the agglomeration of PEDOT:PSS/Ti_3_C_2_/GQD.

Subsequently, the result from Figure 4A shows the improved electrochemical behavior can be attributed to the excellent electrical conductivity of the Ti_3_C_2_ and GQD present on the electrode surface could accelerate the electron transfer rate at the electrode/electrolyte interface [43,44,45]. In regard to the presence of Ti_3_C_2_ and GQD, they provide additional active surface area, thus resulting in better exposure of electroactive sites compared to PEDOT:PSS alone [23,46]. Interestingly, the electroactive surface area of modified SPCE modified PEDOT:PSS/Ti_3_C_2_/GQD exhibited an increase about two-fold, indicating good synergistic effects between GQD, Ti_3_C_2_, and PEDOT:PSS.

The electrocatalytic activity of fabricated biosensor in this study can briefly be described as below, where FAD is reduced to FADH_2_ during glucose oxidation by accepting electrons from the glucose activity mechanism of an enzyme on the electrode:Glucose + GOx (FAD) → Gluconolactone + GOx (FADH_2_)(3)
GOx (FADH_2_) + O_2_ → H_2_O_2_ + GOx (FAD)(4)
H_2_O_2_ → O_2_ + 2H+ + 2e^−^(5)

## 5. Conclusions

A novel nanomaterial for the development of an electrochemical sensor to determine glucose based on PEDOT:PSS/Ti_3_C_2_/GQD has been proposed in this study. The GOx enzyme was immobilized on PEDOT:PSS/Ti_3_C_2_/GQD modified SPCE by a simple electrochemical process. From CV measurement, the direct electrochemistry of GOx on PEDOT:PSS/Ti_3_C_2_/GQD electrode is observed. PEDOT:PSS/Ti_3_C_2_/GQD modified electrodes are proven to offer the efficient detection of glucose with a comparable limitation (LOD) of 65 µM and LOQ of 217 µM, as well as high sensitivity of 21.64 µAmM^−1^cm^−2^. Thus, the PEDOT:PSS/Ti_3_C_2_/GQD-GOx shows remarkable improvement in terms of stability and repeatability. The results for this study demonstrate simple guidelines to improve SPCE based on nanocomposite PEDOT:PSS/Ti_3_C_2_/GQD for the efficient detection of glucose. The excellent performance of the prepared composite film can also be investigated for the immobilization of other redox enzymes or proteins.

## Figures and Tables

**Figure 1 biosensors-11-00267-f001:**
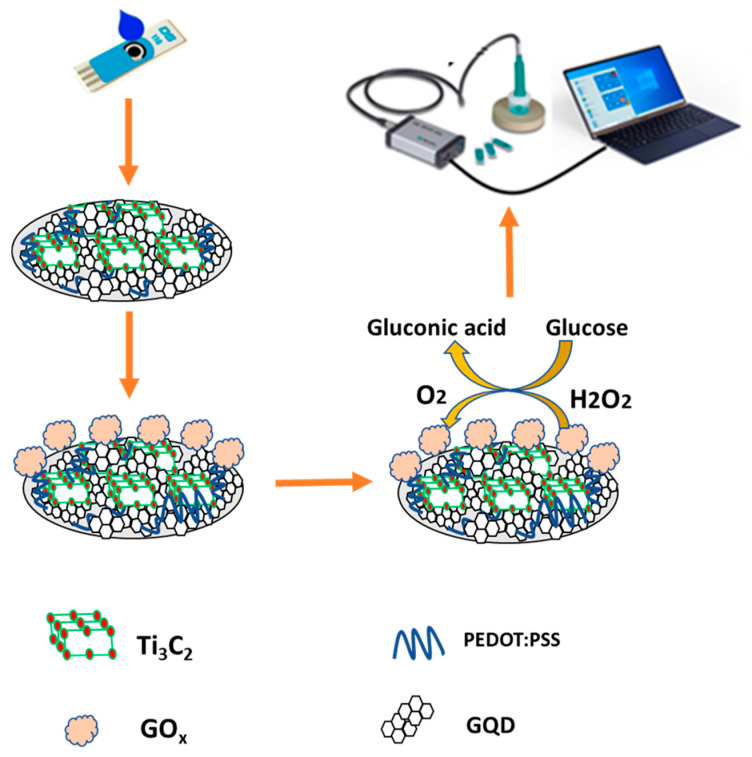
Schematic diagram of fabricated PEDOT:PSS/Ti_3_C_2_/GQD modified SPCE for glucose detection.

**Figure 2 biosensors-11-00267-f002:**
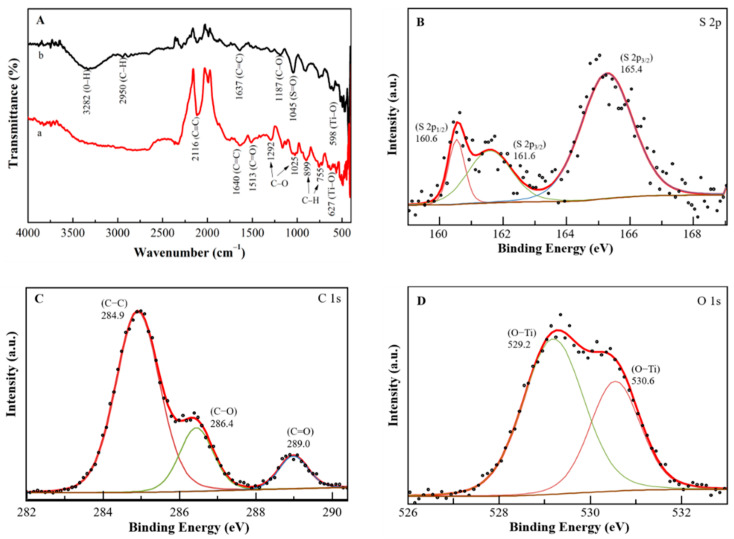
(**A**) FTIR spectrum of (a) PEDOT:PSS/Ti_3_C_2_/GQD and (b) PEDOT:PSS/Ti_3_C_2_/GQD in presence of GOx immobilization. XPS spectrum and high-resolution XPS spectra of (**B**) S 2p, (**C**) C 1s and (**D**) O 1s of the PEDOT:PSS/Ti_3_C_2_/GQD−GOx.

**Figure 3 biosensors-11-00267-f003:**
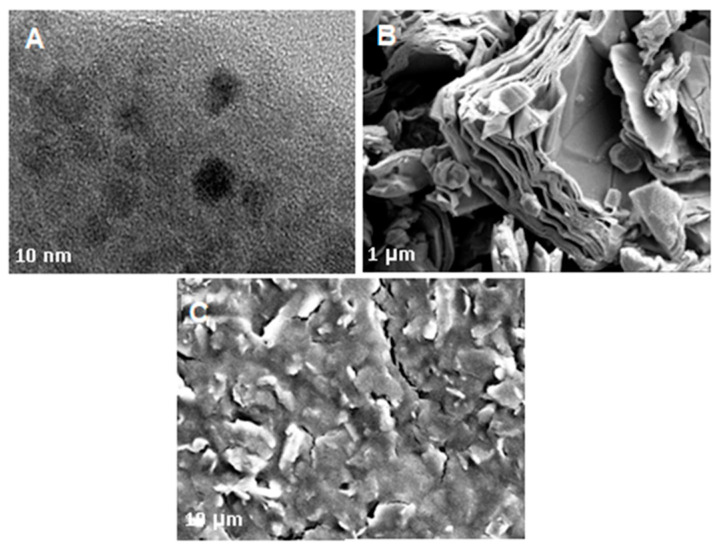
SEM image of (**A**) HRTEM image of GQD. Scale bar corresponds to 20 nm; (**B**) mXene, Ti_3_C_2_, and (**C**) PEDOT:PSS/Ti_3_C_2_.

**Figure 4 biosensors-11-00267-f004:**
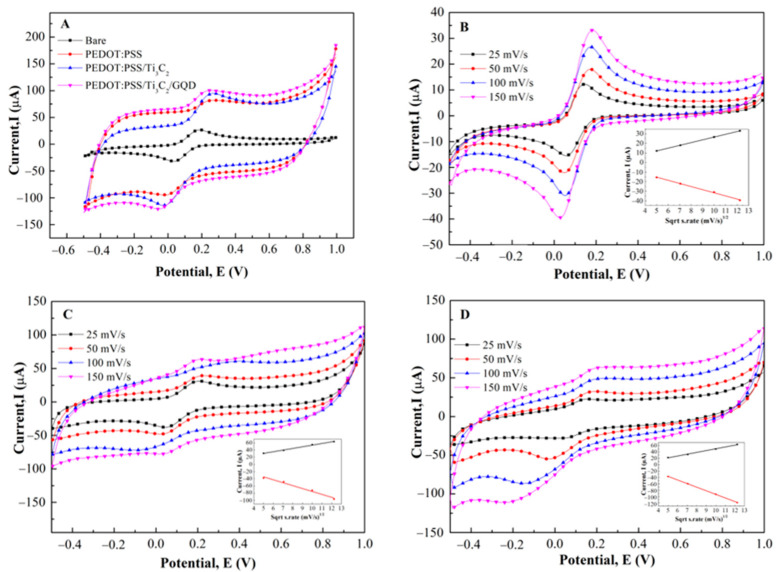
(**A**) CV measurements of SPCE/PEDOT:PSS/Ti_3_C_2_/GQD (Pink) compared to SPCE/PEDOT:PSS (Red), PEDOT:PSS/Ti_3_C_2_ (blue) and bare SPCE (Black) in 0.1 M K_3_[Fe(CN)_6_] with ammonium acetate solution at pH 7. CV measurement of (**B**) Bare SPCE, (**C**) SPCE/PEDOT:PSS/Ti_3_C_2_, and (**D**) SPCE/PEDOT:PSS/Ti_3_C_2_/GQD at various scan rates 25, 50, 100 and 150 mV/s. Anodic Peak currents as a function of square root scan rate for the determination of the effective working surface area (Inset).

**Figure 5 biosensors-11-00267-f005:**
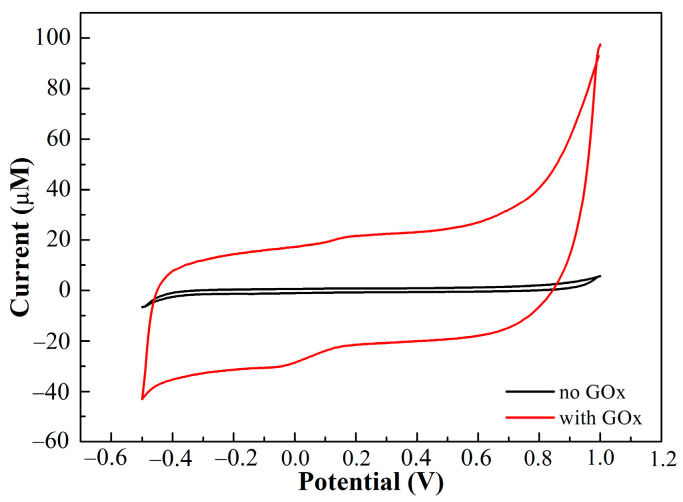
CV response of the SPCE/PEDOT:PSS/Ti_3_C_2_/GQD in the presence and absence of glucose oxidase with 500 μM of glucose in ammonium acetate solution at scan rate of 0.01 V/s.

**Figure 6 biosensors-11-00267-f006:**
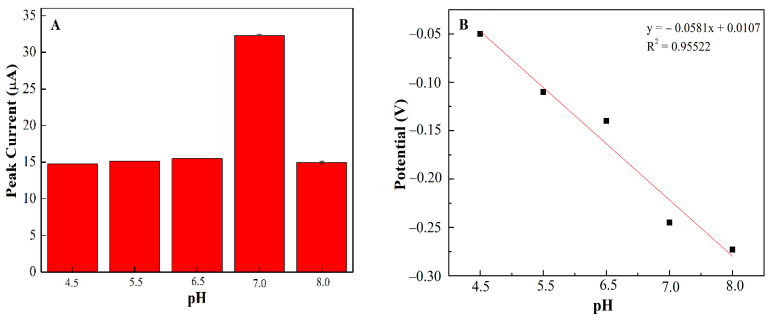
(**A**) The effect of pH on the DPV signal of PEDOT:PSS/Ti_3_C_2_/GQD in 0.1 M ammonium acetate solution; (**B**) The relationship between the potential (V) and pH of solution.

**Figure 7 biosensors-11-00267-f007:**
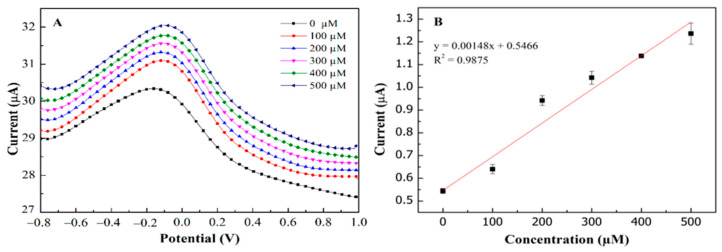
The concentration of glucose detection with linear range 0−500 µM at SPCE/PEDOT: PSS/Ti_3_C_2_/GQD−GOx, (**A**) DPV analysis for 0–500µM of glucose; (**B**) linearity of glucose concentration vs. current oxidation peak (Ipa).

**Figure 8 biosensors-11-00267-f008:**
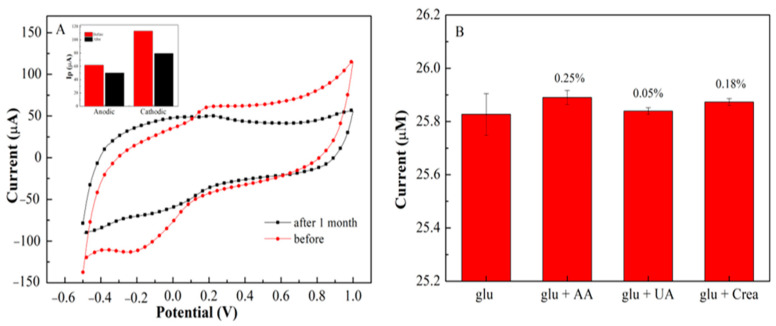
(**A**) CV stability of 1−month storage on PEDOT:PSS/Ti_3_C_2_/GQD/GOx in the prsense 500 µM glucose; (**B**) Interference study of glucose on the PEDOT:PSS/Ti_3_C_2_/GQD−GOx SPCE.

**Table 1 biosensors-11-00267-t001:** Summary of glucose detection in an electrochemical system with modification of SPE.

Modification	Technique	LOD (µM)	Sensitivity (µAmM^−1^cm^−2^)	References
rGO-PEDOT:PSS ^a^	CV, Amperometric	86.8	57.3	[35]
GOx/AuNP/PANI/rGO/NH_2_-MWCNTs ^b^	Amperometric	64.0	246	[36]
Pt/rGO/P3ABA ^c^	Chronoamperometric	44.3	22.0	[37]
Li/rGO/APBA ^d^	CV	30.0	-	[38]
Ni(OH)_2_/AuNp	CV	40.0	-	[39]
GOx/Pt-graphite	Chronoamperometric	10.0	10.5	[40]
PEDOT:PSS/Ti_3_C_2_/GQD-GOx	CV, DPV	65.0	21.64	this work

Glucose detection on modified electrode; ^a^ reduced graphene oxide (rGO); ^b^ multiwalled carbon nanotubes (MWCNTs); ^c^ poly(3-aminobenzoic acid (P_3_ABA); ^d^ aminophenylboronic acid (APBA).

## Data Availability

Data is contained within the article or Appendix A.

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
