# Peer review of "Label Free Glucose Electrochemical Biosensor Based on Poly(3,4-ethylenedioxy thiophene):Polystyrene Sulfonate/Titanium Carbide/Graphene Quantum Dots"

_biosensors, 2021, doi:10.3390/bios11080267_

Round 1
Reviewer 1 Report
The paper presents a electrochemical sensor for the detection of glucose based on GOx supported on Poly(3,4-ethylenedioxy thiophene):Polystyrene Sul-fonate/Titanium Carbide/Graphene Quantum Dots. Although it provides somewhat interesting modification, the manuscritpt should be extensively revised before considering for publication.
Introduction
The last paragraph is not clear to distinguish the authors work to what is found in literature. Please, improve the clarity of this paragraph.
Materials and methods
“ammonium acetate buffer” – An ammonium acetate solution is not a buffer, as the pka from both acetic acid and ammonium hydroxide (4,75 and 9,25, respectively) are distant from the pH 7. The pH value is achieved due the identical magnitude of the hydrolysis reaction from both species. Please correct it at the text.
“electrode consists of the working electrode (carbon), reference electrode (carbon) and 112 counter/auxiliary electrode (Ag/Cl) respectively”. Usually, the reference electrode is the Ag/AgCl and the counter electrode is carbon on screen printed electrodes. Can the authors confirm that information?
What is the role of each layer on the surface?
Why the authors need to store the solution for 24h prior to its utilization? What are the benefits of this procedure?
There are line spacing issues between the section 2.4 and 2.5
Section 3.3 is confusing.Were the cyclic voltammograms displayed in figure 5 obtained in glucose solution with GOx adsorbed on the surface? If that is the case, please rewrite the entire section and improve the caption of figure 5 to accurately describe the data.
The Epa vs pH plot on figure 6 is not directly related to the glucose . The observed change in potential is related to the reduction of hydrogen peroxide produced as subproduct in glucose oxidation by GOx.
The manuscript cites figures 7 and 8 but the figures are absent from the text. Its impossible to verify the data described from both sections 3.5 and 3.6
Author Response
To : Biosensor
Response to Reviewers
Dear Editor,
We appreciate you and the reviewers for your precious time in reviewing our paper and providing valuable comments. It was your valuable and insightful comments that led to possible improvements in the current version. The authors have carefully considered the comments and tried our best to address every one of them. We hope the manuscript after careful revisions meet your high standards. The authors welcome further constructive comments if any. Below we provide the point-by-point responses. All modifications in the manuscript have been mark up using track changes and highlighted in blue color.
Sincerely,
Hazani Zaid, PhD
(Postdoctoral researcher)
Institute of Microengineering and Nanoelectronics (IMEN),
Universiti Kebangsaan Malaysia (UKM), Bangi.
Reviewer 1
The paper presents a electrochemical sensor for the detection of glucose based on GOx supported on Poly(3,4-ethylenedioxy thiophene):Polystyrene Sul-fonate/Titanium Carbide/Graphene Quantum Dots. Although it provides somewhat interesting modification, the manuscript should be extensively revised before considering for publication.
Introduction
- The last paragraph is not clear to distinguish the authors work to what is found in literature. Please, improve the clarity of this paragraph.
Author response.
Thank you for your comment,. we had improved the work as suggested. Please see the line 84-92)
Materials and methods
- ammonium acetate buffer” – An ammonium acetate solution is not a buffer, as the pka from both acetic acid and ammonium hydroxide (4,75 and 9,25, respectively) are distant from the pH 7. The pH value is achieved due the identical magnitude of the hydrolysis reaction from both species. Please correct it at the text.
Author response
Thank you, Indeed, we agree with you. we had changed to ‘ammonium acetate solution’ throughout the paper
- “electrode consists of the working electrode (carbon), reference electrode (carbon) and counter/auxiliary electrode (Ag/Cl) respectively”. Usually, the reference electrode is the Ag/AgCl and the counter electrode is carbon on screen printed electrodes. Can the authors confirm that information?
Author response
Thank you. This has now been corrected. Please see the change in line 111 and 112
- What is the role of each layer on the surface?
Author Response
Briefly, we try to improve the sensitivity of the developed sensor by altering PEDOT:PSS conductivity. Although PEDOT:PSS-high has a good conductivity with high biocompatibility (sloniewska et al., 2020), however there is not peak produce. Threfore, with the additional of GQD and Ti2c3, the result for CV as shown in Fig 5 was obtained instead of PEDOT:PSS alone.
- Why the authors need to store the solution for 24h prior to its utilization? What are the benefits of this procedure?
Author response
Thanks for the comment. 24 hours was crucial to observed their changes any insolubility, stability and homogeneity of nanocomposite prior to the attachment onto electrode surface.
- There are line spacing issues between the section 2.4 and 2.5
Author response
Thanks, for your comment. This has now been corrected as suggested.
- Section 3.3 is confusing. Were the cyclic voltammograms displayed in figure 5 obtained in glucose solution with GOx adsorbed on the surface? If that is the case, please rewrite the entire section and improve the caption of figure 5 to accurately describe the data.
Author response
We thank the reviewer for pointing out this problem. We notice there are confusing issues between caption and word in the text. Yes it true that obtained Fig 5 was occurred in glucose solution with GOx adsorbed on the surface. This has now been amended. Please see the change in line 234
- The Epa vs pH plot on figure 6 is not directly related to the glucose . The observed change in potential is related to the reduction of hydrogen peroxide produced as subproduct in glucose oxidation by GOx.
Author response
Thank you very much for your nice reminder. Please see the change made in (line 264)
- The manuscript cites figures 7 and 8 but the figures are absent from the text. Its impossible to verify the data described from both sections 3.5 and 3.6
Author response
we sorry for the mistake. Its already added to the manuscript
Reviewer 2 Report
This manuscript present an interesting process to prepare a PEDOT:PSS/Ti3C2/GQD modified screen-printed carbon electrode (SPCE) for glucose sensing, which showed improved current and sensitivity in comparison with unmodified SPCE electrodes and may advance diagnoses for potential patients with diabetes.. I have several questions regarding the characterization and performance of this biosensor that should be addressed before the paper can be published.
My specific comments are below:
- What’s the function of Ti3C2 in the composite? Especially, GQD has been added to improve the electrochemical behavior and electrical conductivity. In Fig 4, there is no significant improvement of CV current of PEDOT:PSS/Ti3C2/GQD over PEDOT:PSS alone. So, is it possible to just use PEDOT:PSS for your sensor. What’s the sensitivity then?
- How much enzyme is included in one sensor? Since it is a physical mixture, I assume it will be dissolved in water and only for one-time use, right?
- For all reported potential values in figures, is there any reference electrode? If you have ref, please indicate them in figures. Or the author applied two-electrode scheme for measurement?
- In Fig 6, why there is almost constant current across pH values but a linear decrease in potential? What’s the raw data for these derived data?
- The caption for figure 6 is wrong and Fig. 7 and Fig. 8 are missed, which is a big lapse for the manuscript.
Author Response
To: Biosensor
Response to Reviewers
Dear Editor,
We appreciate you and the reviewers for your precious time in reviewing our paper and providing valuable comments. It was your valuable and insightful comments that led to possible improvements in the current version. The authors have carefully considered the comments and tried our best to address every one of them. We hope the manuscript after careful revisions meet your high standards. The authors welcome further constructive comments if any. Below we provide the point-by-point responses. All modifications in the manuscript have been mark up using track changes and highlighted in blue color
Sincerely,
Hazani Zaid, PhD
(Postdoctoral researcher)
Institute of Microengineering and Nanoelectronics (IMEN),
Universiti Kebangsaan Malaysia (UKM), Bangi.
Comments and Suggestions for Authors (Reviewer 2 )
This manuscript present an interesting process to prepare a PEDOT:PSS/Ti3C2/GQD modified screen-printed carbon electrode (SPCE) for glucose sensing, which showed improved current and sensitivity in comparison with unmodified SPCE electrodes and may advance diagnoses for potential patients with diabetes.. I have several questions regarding the characterization and performance of this biosensor that should be addressed before the paper can be published.
My specific comments are below:
- What’s the function of Ti3C2 in the composite? Especially, GQD has been added to improve the electrochemical behavior and electrical conductivity. In Fig 4, there is no significant improvement of CV current of PEDOT:PSS/Ti3C2/GQD over PEDOT:PSS alone. So, is it possible to just use PEDOT:PSS for your sensor. What’s the sensitivity then?
Author response
Thanks for the question. Yes it is possible just to use PEDOT:PSS alone seem previous study has shown its capability. However, in this study, the purpose of Ti3C2 would be enhance the capability of the fabricated biosensor in the presence of GQD (line 204). Moreover, the advantages of this study based on label free relying on the peak at -0.1V where its has appears after integration with TI2C3 and GQD. Morover better biosensor performance (sensitivity) with the modification of PEDOT:PSS together with Ti3C2 compared to PEDOT:PSS alone.
- How much enzyme is included in one sensor? Since it is a physical mixture, I assume it will be dissolved in water and only for one-time use, right?
Author response
Thanks for the question. The amount of enzyme is already stated in line 120. Yes, it is the single-used electrode. We also add the information regarding this matter. (Line 111)
- For all reported potential values in figures, is there any reference electrode? If you have ref, please indicate them in figures. Or the author applied two-electrode scheme for measurement?
Author response
We would like to point out that we used a screen-printed carbon electrode integrated with working electrode (carbon), reference electrode (Ag/Cl) and counter/auxiliary electrode (carbon).
- In Fig 6, why there is almost constant current across pH values but a linear decrease in potential? What’s the raw data for these derived data?
Author response
Thank you for the question. In diagram A, the raw data of this experiment was examined using DPV to investigated the best pH condition, while in Fig B, the result getting based on the variation of formal potential vs. pH values where it was extracted from cyclic voltammetry.
- The caption for figure 6 is wrong and Fig. 7 and Fig. 8 are missed, which is a big lapse for the manuscript.
Author response
We thank the reviewer for this observation. The problem has been fixed

Round 2
Reviewer 1 Report
Dear authors,
The manuscript has been improved extensively thus I now reccomend your it for publication after minor revision.
There is only a minor correction needed at line spacing of the section 2.4 which is still different from the entire manuscript.
Reviewer 2 Report
No further comments about the manuscript